# The extant immunoglobulin superfamily, member 1 gene results from an ancestral gene duplication in eutherian mammals

**Courtney L. Smith[1], Paul M. Harrison[2], Daniel J. Bernard**[1]*

**1** Department of Pharmacology and Therapeutics, McGill University, Montreal, Quebec, Canada,
**2** Department of Biology, McGill University, Montreal, Quebec, Canada

* daniel.bernard@mcgill.ca

**Data Availability Statement:** All relevant data are within the manuscript and its Supporting Information files.

## Abstract

Immunoglobulin superfamily, member 1 (IGSF1) is a transmembrane glycoprotein with high expression in the mammalian pituitary gland. Mutations in the *IGSF1* gene cause congenital central hypothyroidism in humans. The IGSF1 protein is co-translationally cleaved into N- and C-terminal domains (NTD and CTD), the latter of which is trafficked to the plasma membrane and appears to be the functional portion of the molecule. Though the IGSF1-NTD is retained in the endoplasmic reticulum and has no apparent function, it has a high degree of sequence identity with the IGSF1-CTD and is conserved across mammalian species. Based upon phylogenetic analyses, we propose that the ancestral *IGSF1* gene encoded the IGSF1-CTD, which was duplicated and integrated immediately upstream of itself, yielding a larger protein encompassing the IGSF1-NTD and IGSF1-CTD. The selective pressures favoring the initial gene duplication and subsequent retention of a conserved IGSF1-NTD are unresolved.

## Introduction

Immunoglobulin superfamily, member 1 (IGSF1) deficiency syndrome is a rare disorder (1 in 100,000) of central hypothyroidism, testicular enlargement, hypoprolactinemia, and growth hormone dysregulation caused by loss of function mutations in the X-linked *IGSF1* gene [1–4]. IGSF1, a protein of unknown function, is highly expressed in the mammalian pituitary gland, hypothalamus, testes, and fetal liver [5]. In the pituitary, IGSF1 is produced in thyrotrope, lactotrope, and somatotrope cells, which secrete thyroid-stimulating hormone (TSH), prolactin, and growth hormone, respectively [3, 5]. In the absence of IGSF1, pituitary expression of the thyrotropin-releasing hormone (TRH) receptor and the TSH subunits (*Tshb* and *Cga*) is reduced in mice [6]. TRH-induced TSH secretion is correspondingly impaired in *Igsf1* knockout mice, likely explaining their central hypothyroidism [6].

The full-length *IGSF1* cDNA encodes a 12 immunoglobulin (Ig) loop transmembrane glycoprotein that is co-translationally cleaved at an internal signal sequence in the endoplasmic reticulum (ER) into N-terminal (NTD) and C-terminal domains (CTD) [7] (Fig 1A). Following cleavage, both domains possess Ig loops (five and seven, respectively) in the ER lumen, a

**Funding:** This work was supported by Canadian Institutes of Health Research operating grant MOP-133557 to D.J.B. C.L.S. was partially supported by internal fellowships from the McGill University Faculty of Medicine and Health Sciences. P.M.H is funded by a Discovery grant from the Natural Sciences and Engineering Research Council of Canada.

**Competing interests:** The authors have declared that no competing interests exist.

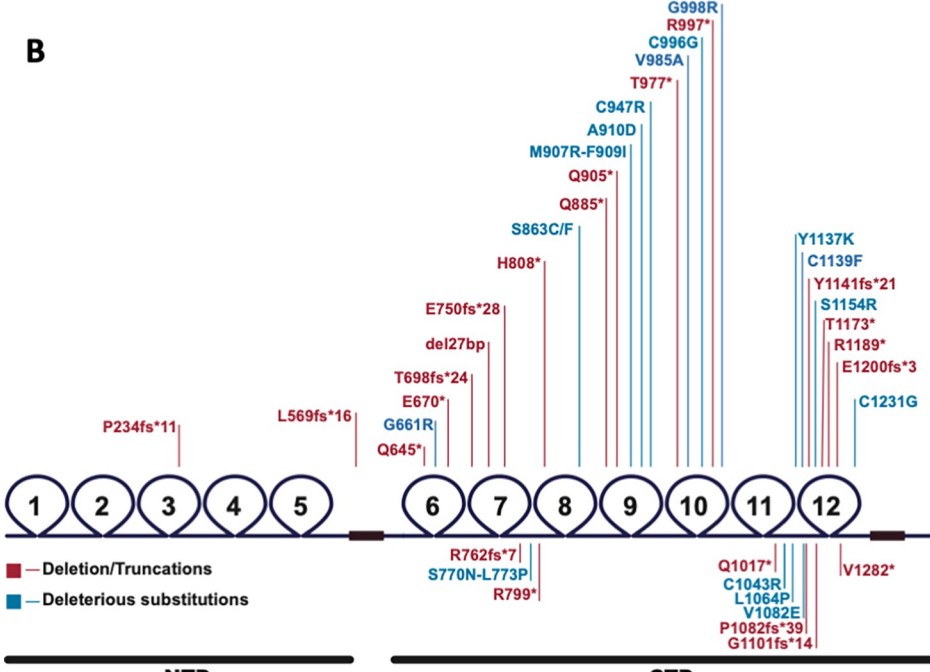

**Fig 1. IGSF1 domain structure and known mutations in *IGSF1*.** A) IGSF1 is co-translationally cleaved into the NTD and CTD by signal peptidase and signal peptide peptidase. The NTD is retained in the ER, while the CTD is trafficked to the plasma membrane. The figure was generated in BioRender. B) Known mutations in *IGSF1* cluster in the CTD coding region.

single transmembrane domain, and a short cytoplasmic carboxy tail. The NTD is retained in the ER, whereas the CTD is trafficked to the plasma membrane and appears to be the functional part of the molecule [7]. This concept is supported by the fact that the vast majority of intragenic mutations in *IGSF1* map to the part of the gene encoding the CTD, and these

mutations inhibit the protein's plasma membrane trafficking (Fig 1B) [3, 8–18]. Two disease-associated mutations in the NTD have been described thus far, but both cause frameshifts that preclude expression of the CTD [4, 8, 9, 11–20].

At least two observations suggest that the extant *IGSF1* gene is the product of gene duplication. First, the NTD and CTD have high sequence identity and domain structure similarity [21]. Second, at least in rodents, the pituitary expresses an mRNA isoform that encodes only the CTD [22]. We hypothesize that the ancestral *IGSF1* gene in mammals encoded the CTD, and that this gene was duplicated during evolution and integrated immediately 5' of itself. To test the idea that the NTD arose from duplication of the CTD, we performed phylogenetic analyses of IGSF1. Next, we compared *IGSF1/Igsf1* cDNA and genomic sequences across a large number of species and explored the conservation of the seemingly non-functional NTD.

## Materials and methods

### Multiple sequence alignment and phylogenetic trees

Orthologs of IGSF1 and its relatives (A1BG, OSCAR, FCAR, TARM1, NCR1, LILRA5, and VSTM1) were found using the bi-directional best hits (BBH) method in *Mammalia* and other amniotes. Organisms that have reference proteomes in the UniProt database (https://www.uniprot.org/proteomes/) were used for ortholog searching [23]. Orthologs of IGSF1 were split into the NTD and CTD parts of the molecules. Multiple sequence alignments of the orthologs, together with the IGSF1-NTD and -CTD regions, were made using Clustal Omega [24], with default parameters. The proteins used in our analyses are collated in S1 Raw images.

Phylogenetic trees were made using PhyML with default parameters [25], except as described in the relevant figure legends. Trees were drawn using FigTree (http://tree.bio.ed.ac.uk/software/figtree/).

### Identity and similarity of Ig loops

Amino acid sequences of individual IGSF1 Ig loops were identified from the human NCBI Reference Sequence: NP_001546.2. Loops from the NTD were compared with loops from the CTD using the NCBI BLASTP tool to determine sequence identity and similarity [26].

### Cell transfection and immunoblotting

HEK293 cells (provided by Dr. Terry Hébert, McGill University, Montréal) were cultured in DMEM (4.5 g L$^{-1}$ glucose, with L-glutamine and sodium pyruvate) containing 5% (v/v) FBS. Cells were seeded in 6-well plates at a density of 600,000 cells/well. The following day, cells were transfected with 2 μg of pcDNA3.0, full-length (IGSF1-1; NM_177591.4) or CTD-only (IGSF1-4; NM_183336.2) murine IGSF1 expression plasmid per well using 6 μg polyethylenimine (1:3 ratio). Twenty-four hours after transfection, total protein lysates were extracted as previously described [6]. For lysates from tissues, pituitary glands from 7–8-week-old male C57BL/6J mice were processed as previously described [6]. Protein concentrations were measured using the Pierce BCA protein assay kit (23227, ThermoFisher Scientific) following the manufacturer's instructions. Protein lysates from cells (20 μg) and pituitary (30 μg) were denatured at 72°C for 15 minutes, cooled on ice for 5 minutes, and then deglycosylated with 500 U PNGase F (P0704S, New England BioLabs) or left untreated for 2 hours at 37°C prior to being resolved by sodium dodecyl sulfate–polyacrylamide gel electrophoresis (SDS-PAGE) on an 8% (v/v) tris-glycine gel and transferred to a nitrocellulose membrane (10600001, GE Healthcare). The rabbit anti-IGSF1-CTD antibody (1:1000, RRID:AB2631165) was previously described [7]. The membrane was washed and incubated in goat anti-rabbit IgG (H+L)-HRP conjugate

(1706515, Bio-Rad), washed, and incubated with ECL (NEL103E001EA, PerkinElmer) before imaging.

All animal work was performed in accordance with institutional and federal guidelines under an animal use protocol (AUP5204) approved by the McGill University Facility Animal Care Committee DOW-A.

## Results

### Sequence conservation between the IGSF1-NTD and -CTD

Examination of the amino acid sequence of human IGSF1 revealed remarkable identity between its NTD and CTD. Ig loops from the NTD clustered with Ig loops from the CTD in a phylogenetic tree–loop 1 with 6, loop 2 with 7, loop 3 with 9 and 11, loop 4 with 10, and loop 5 with 12, with branch support values of 1, 0.91, 0.89, 0.96, and 0.98, respectively (Fig 2). The amino acid sequence identity between the Ig loops that clustered together ranged from 44 to 63%, and the similarity ranged from 58–80% (Table 1). In contrast, non-clustering Ig loops had sequence identities from 23 to 44%, and similarities from 41 to 60%. Ig loop 8 in the CTD lacked a corresponding loop in the NTD (Fig 2C). Further, the branch lengths for the NTD Ig loops 2, 3, 4 and 5 were longer than for the CTD Ig loops, indicating less constraint on their sequences.

### IGSF1-NTD and -CTD evolved independently

The similarity between the IGSF1-NTD and -CTD suggested that the duplication of one may have led to the emergence of the other. We performed another phylogenetic analysis, this time examining the NTD and CTD sequences across species. We found no evidence of IGSF1 in non-mammalian species and we did not observe clustering with sequences from non-eutherian amniotes. The CTDs in mammals clustered together, as did the NTDs (Fig 3). However, the NTDs and CTDs clustered separately in the tree (compare IGSF1-N and IGSF1-C in Fig 3).

### IGSF1 and the LRC family

As IGSF1 was recently designated a member of the leukocyte receptor cluster (LRC) family [21], we included LRC homologs of IGSF1 in our analysis. The human IGSF1-NTD and -CTD clustered with sets of orthologs of the alpha-1-B glycoprotein (A1BG) (Fig 3). BBH orthologs of IGSF1-NTD and -CTD, as well as for A1BG, were observed only in eutheria, indicating a common ancestry during the early eutherian epoch. These proteins did not cluster with LRC homologs osteoclast-associated immunoglobulin-like receptor (OSCAR), immunoglobulin alpha Fc receptor (FCAR), T-cell-interacting, activating receptor on myeloid cells protein 1 (TARM1), natural cytotoxicity triggering receptor 1 (NCR1), leukocyte immunoglobulin-like receptor A5 (LILRA5), or V-set and transmembrane domain-containing protein 1 (VSTM1), indicating distinct ancestry. VSTM1 contains only one Ig loop, and this loop was aligned to IGSF1-NTD Ig loop 4 (loop 10 in the CTD) by Clustal Omega [24].

We performed an additional phylogenetic analysis using only the sequence encoding Ig loop 4 of IGSF1, which is the deepest part of the Clustal Omega alignment, comprising all of the LRC sequences examined (Figs 4 and 5). Here again, despite the limited sequence information, IGSF1-NTD and -CTD clustered with A1BG, albeit with a weaker branch support value of 0.6 (Fig 4). In this tree, among the other LRC members examined, OSCAR clustered most closely with IGSF1-NTD and -CTD and A1BG (Fig 4).

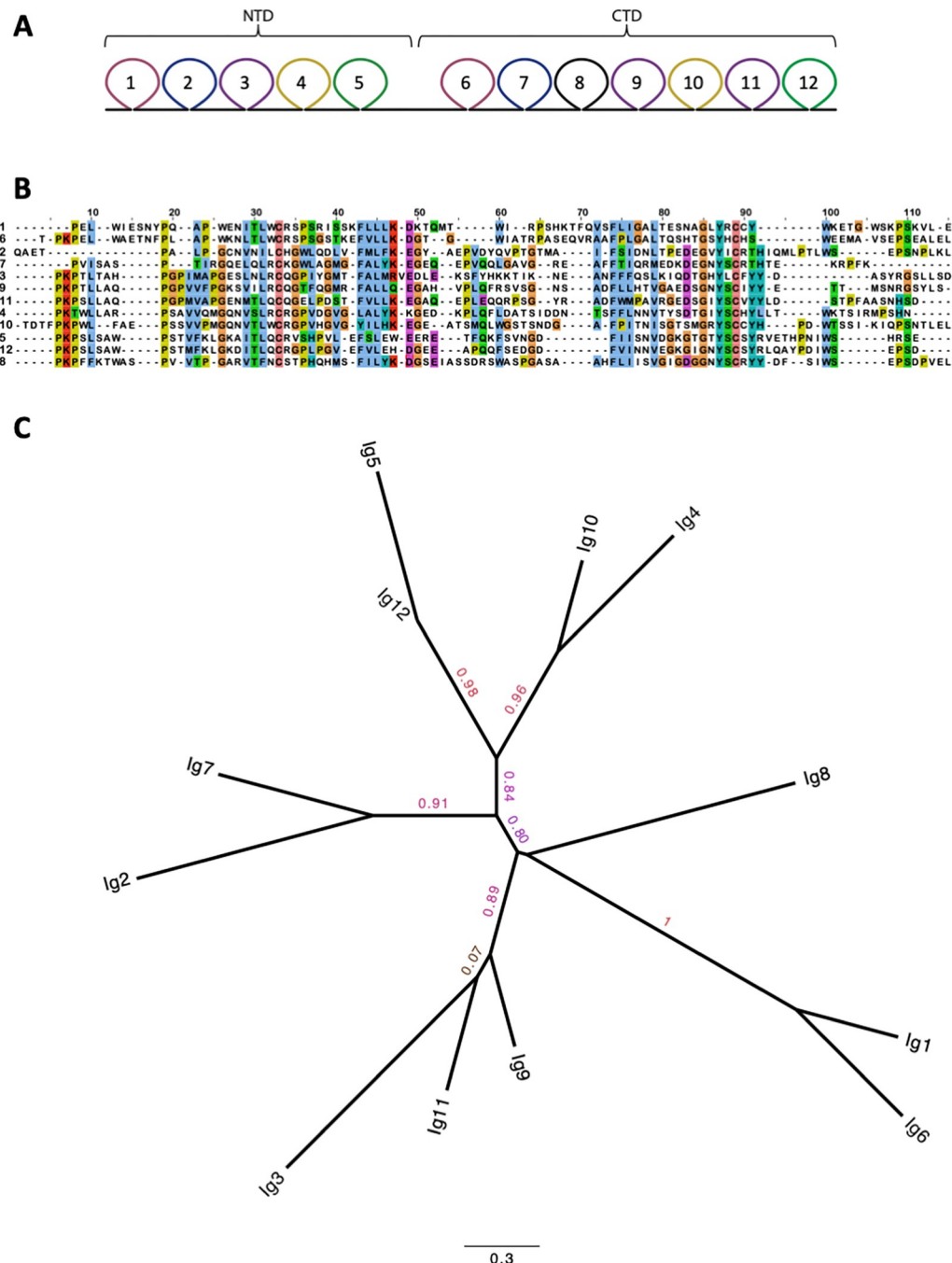

**Fig 2. Ig loops in the NTD cluster with specific loops from the CTD.** A) Ig loops with the same colour have the highest sequence identity. B) Sequence alignment of each Ig loop with highly conserved residues highlighted (blue = hydrophobic, red = positive charge, magenta = negative charge, green = polar, pink = cysteine, orange = glycine, yellow = proline, cyan = aromatic, non-conserved = white). Ig loops are ordered based on sequence identity; Ig1 with Ig6, Ig2 with Ig7, and so on. Sequences are from human accession number Q8N6C5. C) A phylogenetic tree of the 12 Ig domains in human IGSF1. The branches are labelled with PhyML aLRT support values.

**Table 1. Identity and similarity between the Ig loops in human IGSF1 NTD and CTD.**

| CTD loop | 6 | 7 | 8 | 9 | 10 | 11 | 12 |
|---|---|---|---|---|---|---|---|
| NTD loop | Identity | | | | | | |
| 1 | **34/71 (48%)** | 17/59(29%) | 23/75(31%) | 10/43(23%) | 22/75(29%) | 17/61(28%) | 23/74(31%) |
| 2 | 29/96(30%) | **36/82(44%)** | 32/95(34%) | 23/70(33%) | 27/93(29%) | 32/91(35%) | 29/91(32%) |
| 3 | 21/76(28%) | 24/70(34%) | 29/96(30%) | **32/74(43%)** | 21/74(28%) | 34/92(37%) | 22/78(28%) |
| 4 | 22/74(30%) | 24/58(41%) | 29/95(31%) | 33/75(44%) | **46/94(49%)** | 26/75(35%) | 33/93(35%) |
| 5 | 24/74(32%) | 31/90(34%) | 27/94(29%) | 26/72(36%) | 31/91(34%) | 29/89(33%) | **55/87(63%)** |
| | Similarity | | | | | | |
| 1 | **48/71(67%)** | 25/59(42%) | 38/75(50%) | 19/43(44%) | 36/75(48%) | 32/61(52%) | 32/74(43%) |
| 2 | 40/96(41%) | **48/82(58%)** | 47/95(49%) | 33/70(47%) | 43/93(46%) | 46/91(50%) | 43/91(47%) |
| 3 | 33/76(43%) | 37/70(52%) | 45/96(46%) | **45/74(60%)** | 37/74(50%) | **56/92(60%)** | 42/78(53%) |
| 4 | 29/74(39%) | 34/58(58%) | 39/95(41%) | 45/75(60%) | **62/94(65%)** | 41/75(54%) | 49/93(52%) |
| 5 | 29/74(39%) | 45/90(50%) | 39/94(41%) | 40/72(55%) | 45/91(49%) | 44/89(49%) | **70/87(80%)** |

BLASTp analysis of loops from the human NTD compared to loops from the human CTD. Loops with greatest identity and similarity are bolded.

### Distinct *Igsf1* transcripts encode the same cell surface protein

Multiple *IGSF1/Igsf1* mRNA isoforms have been characterized in human and rodent pituitary glands [22, 28, 29]. The murine *Igsf1* isoform 1 (IGSF1-1) transcript encodes the full-length IGSF1 protein, including both the NTD and CTD. In contrast, murine *Igsf1* isoform 4 (IGSF1-4) transcript, which initiates in intron 9, encodes the CTD alone. When expressed in heterologous HEK293 cells, the CTDs from IGSF1-1 and IGSF1-4 were indistinguishable and co-migrated with IGSF1 endogenously expressed in the murine pituitary (Fig 6).

### Discussion

According to our analysis, a gene duplication event gave rise to the extant *IGSF1* gene early in eutherian evolution since their last common ancestor. We propose that the ancestral *IGSF1* gene encoded the IGSF1 C-terminal domain (CTD) alone. This gene was then duplicated and inserted immediately upstream (5') of itself on Chr. X, creating a novel gene that encodes a large transcript from which the N-terminal domain (NTD) and CTD are co-translationally derived. Consistent with our argument, most of the Ig loops in the NTD and CTD cluster together in a phylogenetic tree (1 and 6; 2 and 7; 3 with 9 and 11; 4 and 10; and 5 and 12). Ig loop 8 in the CTD does not cluster with other loops, suggesting that it was lost from the NTD during or after the duplication event. Ig loop 3 of the NTD shares a root with loops 9 and 11 of the CTD, with a high branch support value of 0.89. We suggest two possibilities for this similarity: 1) during the gene duplication, Ig loop 11 was additionally duplicated and inserted upstream to create Ig loop 9 (or vice versa, Ig loop 9 was duplicated and inserted downstream of itself), or 2) Ig loop 9 or 11 was earlier duplicated and loop 11 was subsequently lost from the NTD during duplication of the CTD (Fig 7). Ig loop 4 shares high sequence similarity with loops 9 and 10, though it is more similar to, and clusters with, loop 10. Additionally, loops 4/10 and 5/12 have higher similarity than other loops clustered together in the tree. This sequence similarity could indicate that these loops are particularly important for protein function. Interestingly, a high proportion of pathogenic mutations cluster in and around loops 10 and 12 (Fig 1B).

Several observations converge to support our argument that the CTD-, rather than NTD-encoding part of *IGSF1* was duplicated. First, the CTD is more complex (7 Ig loops) than the NTD (5 Ig loops) and is more highly conserved. As can be seen in the phylogram comparing

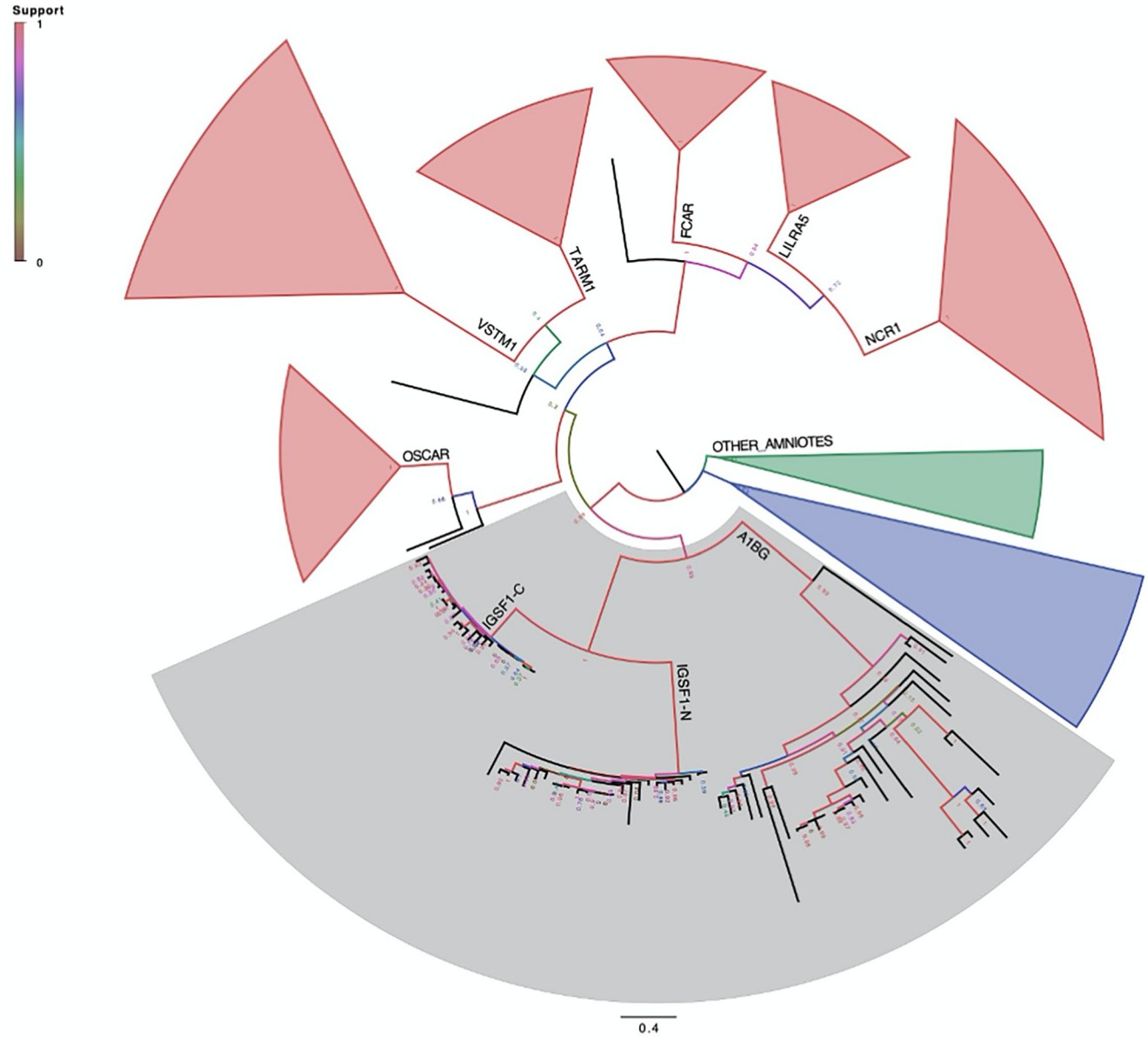

**Fig 3. IGSF1 has high sequence similarity in all mammals and only occurs in chordates.** Phylogenetic tree based on a Clustal Omega alignment of IGSF1 and its human paralogs and their sets of orthologs across eutheria made using PhyML default parameters, except that starting tree topology and invariable sites were optimized, and the PhyML option for using the best of the available tree searching operations was applied. Bi-directional best hits (BBH) from a selection of non-eutherian amniotes were included as outgroups. The IGSF1-NTD and -CTD regions were aligned separately. The clades for IGSF1-NTD and -CTD, A1BG, OSCAR, FCAR, TARM1, NCR1, LILRA5, and VSTM1, and 'Other Amniote' sequences are labelled on the appropriate branches, and collapsed to wedge shapes, except for the IGSF1 and A1BG clades, which are highlighted in grey. The nodes at which branches end were color-coded by PHyML aLRT branch support value as indicated by the heat map. The tree was drawn using FigTree.

Ig loops from the NTD and CTD (Fig 2C), branch lengths are longer for all but one of the Ig domains in the NTD compared to the CTD, indicating that the NTD is mutating at a faster rate at the amino acid level (similarly in Fig 3). In fact, the CTD is mutating more slowly than any member of the LRC family (Figs 3 and 4). Second, at least in rodents, there is a transcript that encodes the CTD alone (what we previously referred to as isoform 4). Transcription of isoform 4 initiates in intron 9 [22]. The open reading frame of the resulting mRNA contains a signal peptide coding sequence at its N-terminus. This signal peptide enables the CTD derived

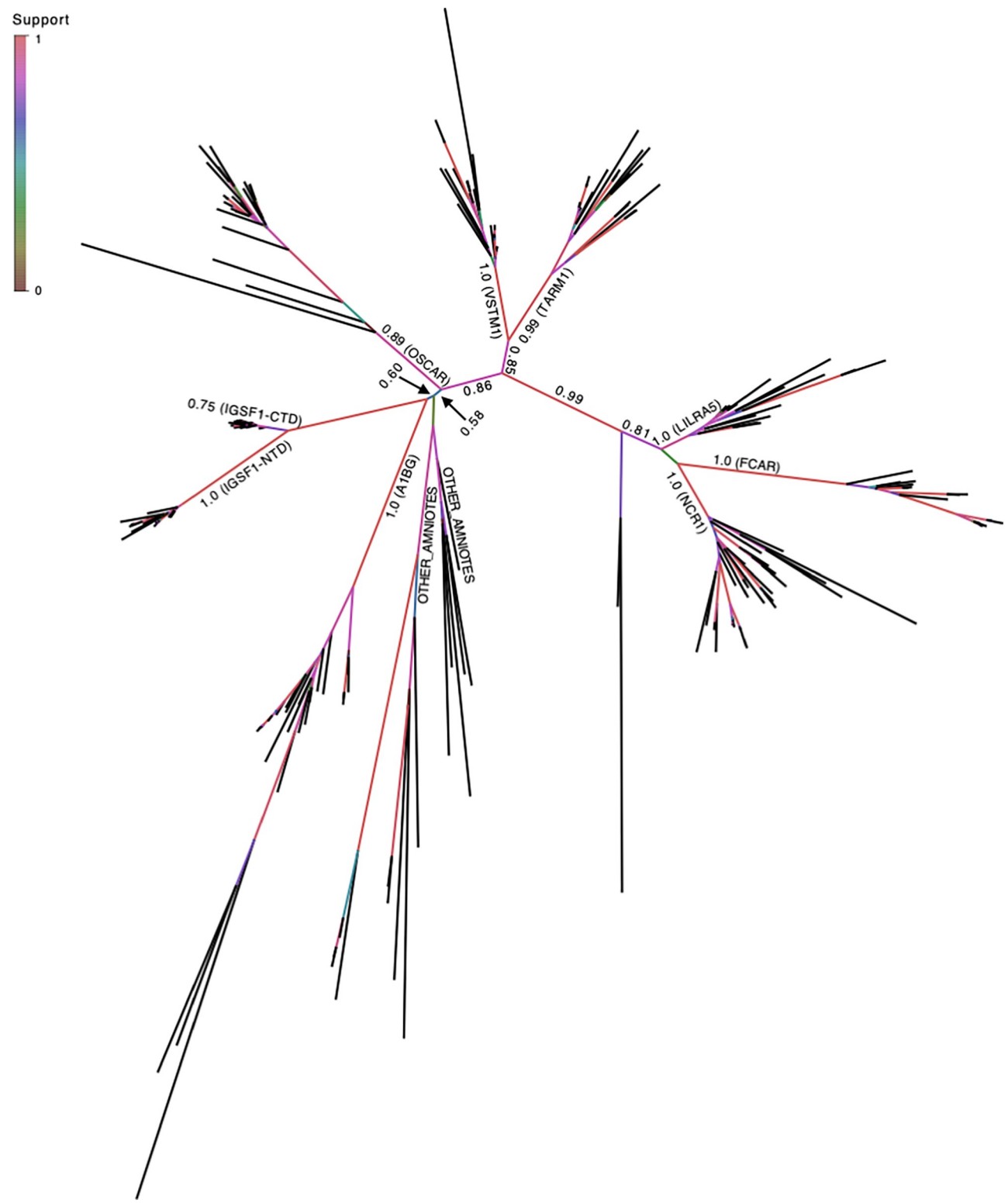

**Fig 4. IGSF1 loop 4 is a highly conserved member of the LRC family.** Phylogenetic tree based on a Clustal Omega alignment of IGSF1 Ig loop 4 and its human paralogs and their sets of orthologs across eutheria. In addition, BBH hits from a selection of non-eutherian amniotes were included as outgroups. The clades for IGSF1-NTD and -CTD regions, A1BG, OSCAR, FCAR, TARM1, NCR1, LILRA5, and VSTM1, and 'Other Amniote' sequences are labelled on the appropriate branches. The nodes at which branches end are labelled and colour-coded by PHyML aLRT branch support value as indicated by the heat map.

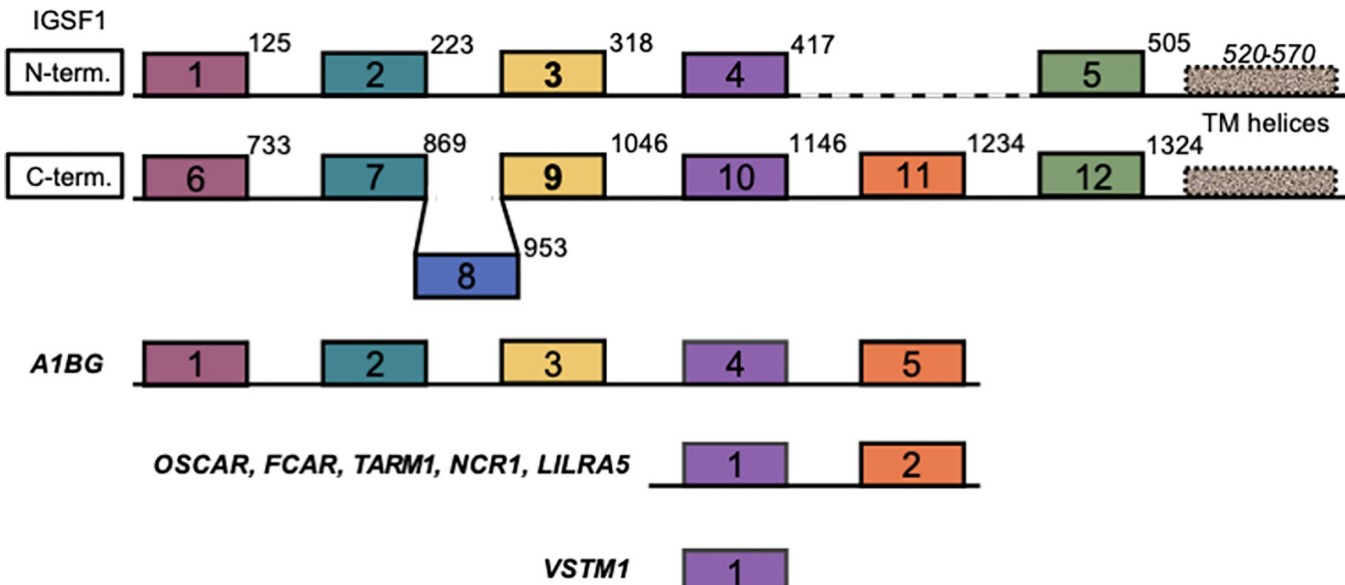

**Fig 5. IGSF1 Ig loops cluster with specific Ig loops from other LRC members.** A schematic depiction of the Ig-like domains in IGSF1-NTD and -CTD, A1BG, OSCAR, FCAR, TARM1, NCR1, LILRA5, and VSTM1. These domain alignments were taken from the Clustal Omega multiple sequence alignment. The endpoints of the domains from Pfam [or SMART, if there was no Pfam annotation; these are both taken from InterPro [27]] are labelled in italics.

from isoform 4 to be expressed as a plasma membrane protein (Fig 6). It also provides the basis for the internal cleavage of the full-length protein into the NTD and CTD by signal peptidase [7, 22] (Fig 1A). We are unaware of other cellular proteins that are processed in this manner (i.e., via an internal signal peptide). The most parsimonious explanation is that this signal peptide has maintained its ancestral function.

Interestingly, the human equivalent of murine *Igsf1* isoform 4 has not been reported. This suggests that transcription from what we propose to be the ancestral promoter in intron 9 has been lost in human evolution. Though present, isoform 4 is expressed at far lower levels than the full-length *Igsf1* mRNA (isoform 1) in murine pituitary [22], further suggesting that the activity of this promoter may similarly be diminishing in rodents. Notably, *Igsf1* knockout mice lacking exon 1 express isoform 4, but not isoform 1. Expression levels of isoform 4 in these mice are unaltered relative to wild-type and they do not express the CTD at sufficient levels to functionally compensate for the loss of the CTD derived from isoform 1 [3, 6].

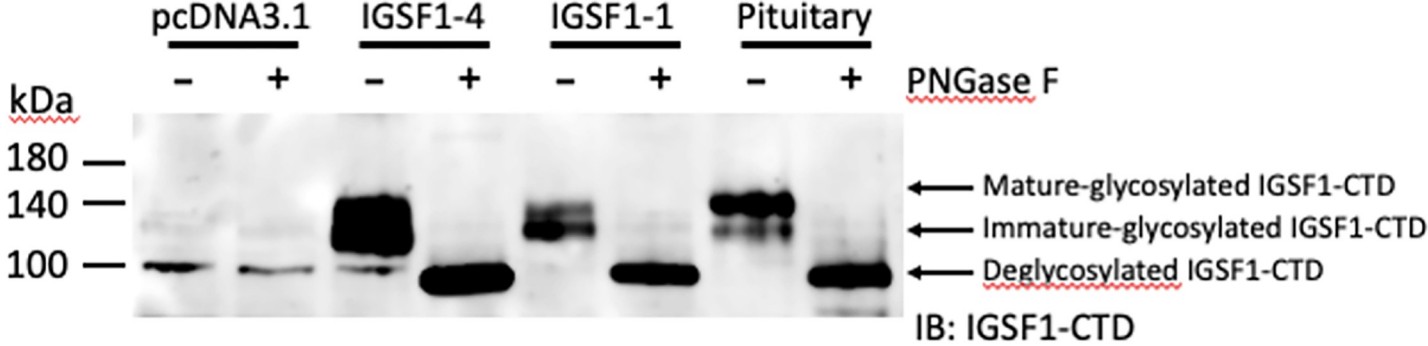

**Fig 6. Full-length and CTD-only *Igsf1* isoforms encode the same protein.** Protein lysates from HEK293 cells transfected with pcDNA3.0, murine full-length IGSF1-1 or murine CTD-coding IGSF1-4 expression vectors, or from murine pituitaries were treated with PNGase F (+) or left untreated (-) and immunoblotted using an IGSF1-CTD antibody. Arrows indicate the mature and immature glycoforms, as well as deglycosylated IGSF1.

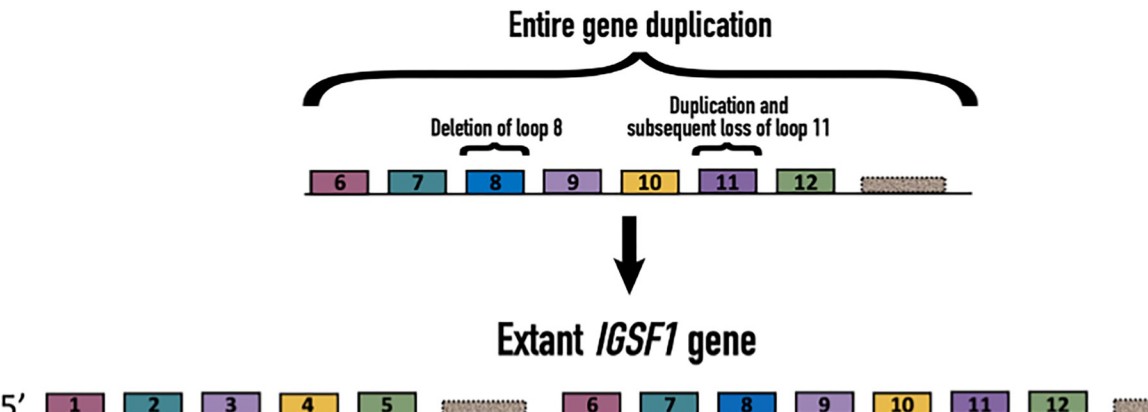

**Fig 7. The extant *IGSF1* gene results from duplication of the ancestral *IGSF1* gene.** A schematic depiction of the evolution of the *IGSF1* gene. The ancestral gene encoded only the IGSF1-CTD. The region encoding Ig loop 8 in the CTD was lost in the NTD. Additionally, loop 11 in the CTD was duplicated and inserted upstream of itself creating loop 9 (or vice versa). Loop 9 was duplicated and emerged in the NTD as loop 3. Loop 11 was lost in the NTD.

Collectively, the data suggest that, over time, activity of the ancestral promoter (in intron 9) has been reduced or lost, with transcription initiating principally from exon 1 and the CTD being derived from the co-translational cleavage of the large precursor protein via the ancestral (internal) signal peptide. Perhaps the acquisition of a new promoter (upstream of exon 1) following the duplication event conferred a selective advantage by quantitatively or qualitatively (spatially or temporally) altering IGSF1 gene and protein expression. In this event, the acquisition of the NTD protein could be coincidental rather than advantageous.

Consistent with this latter idea, the NTD is retained in the ER and has no apparent cellular function [7]. As we show here (Fig 6), the NTD is not needed for the expression or plasma membrane trafficking of the CTD. Given that the protein encoded by the full-length mRNA (isoform 1) is co-translationally cleaved, the NTD at best may serve as a large, but functionless N-terminal prodomain for the CTD. Still, if this is the case, why is the NTD sequence highly conserved? This conservation suggests that the NTD may currently play or perhaps previously played a functional role, even if retained in the ER. That said, the only mutations we are aware of in the NTD that are associated with IGSF1 deficiency (or any disorder) cause frameshifts that preclude expression of the CTD (Fig 1B). Therefore, while there may be selective pressure to maintain the open reading frame of mRNA isoform 1, it is unclear why missense mutations in the NTD-encoding part of the gene are not more prevalent if the NTD is truly functionless.

Finally, as IGSF1 was recently found to be a member of the LRC family (21), we included LRC members in our phylogenetic analysis. We see some divergence between the family members. IGSF1 clusters with A1BG, with only BBH orthologs detected within eutheria, indicating common ancestry during eutherian evolution. It is possible that IGSF1 function, which is not currently known, is more similar to A1BG than to other LRC members. The next most similar protein in the evolutionary analysis is OSCAR (Figs 3 and 4). A1BG and OSCAR function as receptors for extracellular ligands, CRISP-3 and collagens I and III, respectively [30, 31]. We therefore postulate that IGSF1 may similarly function as a receptor or binding protein for one or more extracellular ligands.

## Conclusion

We contend that the ancestral *IGSF1/Igsf1* gene encoded the IGSF1-CTD. During early eutherian evolution, the gene was duplicated and inserted immediately upstream of itself. The novel

gene encodes a large transcript from which two proteins are derived: the IGSF1-NTD and IGSF1-CTD. As the IGSF1-CTDs from the extant and ancestral genes are likely the same (or highly similar) and the IGSF1-NTD is retained in the ER, the adaptive significance of the original gene duplication is not clear but may relate to the acquisition of novel patterns of gene expression. The conservation of the IGSF1-NTD suggests that it may have played an important role that it subsequently lost or that it has a currently unappreciated function in the ER.

## Supporting information

**S1 Raw images.**
(PDF)

## Author Contributions

**Conceptualization:** Daniel J. Bernard.

**Data curation:** Paul M. Harrison.

**Formal analysis:** Courtney L. Smith, Paul M. Harrison, Daniel J. Bernard.

**Funding acquisition:** Paul M. Harrison, Daniel J. Bernard.

**Investigation:** Courtney L. Smith, Paul M. Harrison, Daniel J. Bernard.

**Methodology:** Courtney L. Smith, Paul M. Harrison, Daniel J. Bernard.

**Project administration:** Paul M. Harrison, Daniel J. Bernard.

**Resources:** Paul M. Harrison, Daniel J. Bernard.

**Supervision:** Daniel J. Bernard.

**Validation:** Paul M. Harrison.

**Visualization:** Courtney L. Smith, Paul M. Harrison.

**Writing – original draft:** Courtney L. Smith, Paul M. Harrison, Daniel J. Bernard.

**Writing – review & editing:** Courtney L. Smith, Paul M. Harrison, Daniel J. Bernard.

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
