## [Decision Letter · Decision Letter 0]

14 Apr 2022

The extant immunoglobulin superfamily, member 1 gene results from an ancestral gene duplication in eutherian mammals

PONE-D-22-07485

Dear Dr. Bernard,

We’re pleased to inform you that your manuscript has been judged scientifically suitable for publication and will be formally accepted for publication once it meets all outstanding technical requirements.

Kind regards,

Klaus Roemer

Academic Editor

PLOS ONE

Additional Editor Comments (optional):

Reviewers' comments:

Reviewer's Responses to Questions

**Comments to the Author**

1. Is the manuscript technically sound, and do the data support the conclusions?

Reviewer #1: Yes

2. Has the statistical analysis been performed appropriately and rigorously? 

Reviewer #1: N/A

3. Have the authors made all data underlying the findings in their manuscript fully available?

Reviewer #1: Yes

4. Is the manuscript presented in an intelligible fashion and written in standard English?

Reviewer #1: Yes

5. Review Comments to the Author

Reviewer #1: Smith et al. demonstrated the evolution of the IGSF1 gene through phylogenetic analyzes. They suggested that IGSF1 forms its N-terminal portion over time by duplication and self-integration of its C-terminal portion. Such a hypothesis was established based on the high degree of sequence identity between NTD and CTD and the highly conservation of NTD among mammals. The authors described methods that they used extensively and discussed the results adequately. The analysis and evidences presented while revealing this are sufficient and quite interesting. The manuscript is well written and can make valuable contributions to the literature. Finally, I think this manuscript has enough priority to be published in this journal.

6. PLOS authors have the option to publish the peer review history of their article (what does this mean?). If published, this will include your full peer review and any attached files.

Reviewer #1: No

---

## [Editor Report · Acceptance letter]

11 May 2022

PONE-D-22-07485 

The extant immunoglobulin superfamily, member 1 gene results from an ancestral gene duplication in eutherian mammals 

Dear Dr. Bernard:

I'm pleased to inform you that your manuscript has been deemed suitable for publication in PLOS ONE. Congratulations! Your manuscript is now with our production department. 

Kind regards, 

on behalf of

Dr. Klaus Roemer 

Academic Editor

PLOS ONE